# The Effect of Trace Oxygen Addition on the Interface Behavior of Low-Alloy Steel

**DOI:** 10.3390/ma15041592

**Published:** 2022-02-20

**Authors:** Vlastimil Novák, Lenka Řeháčková, Petra Váňová, Michal Sniegoň, Dalibor Matýsek, Kateřina Konečná, Bedřich Smetana, Silvie Rosypalová, Markéta Tkadlečková, Ľubomíra Drozdová, Petr Klus

**Affiliations:** 1Faculty of Materials Science and Technology, VŠB-Technical University of Ostrava, 17. Listopadu 15, Poruba, 708 00 Ostrava, Czech Republic; lenka.rehackova@vsb.cz (L.Ř.); petra.vanova@vsb.cz (P.V.); michal.sniegon@vsb.cz (M.S.); katerina.konecna@vsb.cz (K.K.); bedrich.smetana@vsb.cz (B.S.); silvie.rosypalova@vsb.cz (S.R.); marketa.tkadleckova@vsb.cz (M.T.); lubomira.drozdova@vsb.cz (Ľ.D.); 2Faculty of Mining and Geology, VŠB-Technical University of Ostrava, 17. Listopadu 15, Poruba, 708 00 Ostrava, Czech Republic; dalibor.matysek@vsb.cz; 3Třinecké Železárny, a.s., Strategy & Innovations, Průmyslová 1000, Staré Mesto, 739 61 Třinec, Czech Republic; petr.klus2@trz.cz

**Keywords:** low-alloy steel, oxygen, surface tension, wetting angle, liquidus temperature, phase interface

## Abstract

This work aims to assess the effect of an oxygen content graded in minimal quantities, on the order of hundreds of ppms, on the determination of surface tension of low-alloy FeCOCr and FeCONi steels in contact with a corundum substrate. Oxygen, as a surface-active element, was segregated at the surface where it interacted with the major components of the alloys, leading to a reduction in surface tension. The sessile drop method was used for wetting tests in the temperature range from steel liquidus temperatures to 1600 °C under nonoxidizing conditions. The effect of oxygen on surface tension and wetting angles was verified by statistical analysis using the Kruskal–Wallis test, which supported the results stating that the values of these quantities decreased with increasing oxygen content. Furthermore, liquidus temperatures, which are of practical importance, were determined by the optical and DTA methods and then compared with theoretically calculated temperature values. It turned out that the increased chromium content causes difficulties in determining surface tension up to 1550 °C due to the formation of a thin Cr_2_O_3_ layer. In addition, SEM and XRD analyses accompanied by calculations in the FactSage oxide database were performed to better understand the wetting mechanism.

## 1. Introduction

Low-alloy steels belong to the group of ferrous materials containing alloying elements in quantities less than stainless steels, that is, less than 10% and more than approximately 2%. These are a wide variety of steel grades with specific challenges in their processing and applications. Therefore, detailed knowledge of their metallurgical, thermodynamic, rheological, and surface properties is in high demand. Of particular importance are those containing mainly nickel and chromium or other alloying elements such as titanium, niobium, vanadium, molybdenum, and others. Their mechanical characteristics and resistance to rust are superior to plain carbon steels [1,2]. However, they also excel in other qualities, such as creep-resistant level and impact/toughness resistance [3,4]. Specifically, they are expected to have high tensile strength and temperature resistance, good corrosion resistance, and fracture toughness in the application domain. Low-alloy steels find applications in the nuclear industry as cooling system components and the in aerospace and automotive industries in the manufacture of gears, crankshafts, and landing gears [5,6,7].

Surface properties, e.g., surface tension and wetting angle, are among the steel characteristics required in steelmaking. Surface tension is a physical force pointing to bulk atoms being pulled and pushed equally in all directions by their neighbors, producing zero net force, unlike atoms on the surface lacking pulling forces in the upward direction and therefore being subjected to a net inward force. In terms of a multi-component alloy, those atoms whose energy state is least affected by the surface will segregate to the liquid surface, where they influence surface tension. Chromium, nickel, and oxygen can segregate on alloy surfaces and influence their surface tension to varying degrees [8,9,10,11,12].

Oxygen is highly surface-active, as its energy changes concerning segregation are relatively high, and thus it influences surface tension on a larger scale [9,11]. Several researchers and research groups have studied the effect of oxygen on iron surface tension. In the case of molten iron, the oxygen partial pressure around 0.01 Pa decreased the surface tension by several percent and altered the course of surface tension temperature dependence to a boomerang-like shape, with a kink around 2150 K due to desorption of oxygen at higher temperatures [13]. Morohoshi et al. studied surface tension depending on oxygen activity using a gas–liquid equilibrium method to control the oxygen activity of liquid iron and keep the carbon activity at a low level. This dependence was well described using the model suggested by Szyszkowski [14]. Kasama [15] found the sigmoidal dependence of surface tension on the logarithm of oxygen partial pressure. According to Brooks, only around 50 ppm of oxygen can reduce the surface tension of pure iron by 50% [16]. In the case of more complex systems, i.e., binary, ternary, quaternary systems, and Fe-based alloys, oxygen also affects the surface tension, lowering it [10,17,18,19,20,21,22]. It is also worth noting that for polycomponent steels, the interface wetting is challenging to simulate, and therefore accurate data are obtained mostly experimentally [10].

During steel processing, the presence of oxygen in the steel matrix is still recognized as one of the main problems. Oxygen can be present in steel in the form of nonmetallic inclusions or as microporosity, and as such, it influences impact and fracture toughness. The oxygen content is affected by the way in which the precursor for steelmaking is stored, as the surface oxidation of the powder is the leading cause by which oxygen interferes. However, the content is also influenced by the very processing method, be it is casting, forging, or hot isostatic pressing, and these can affect the content in the final material by order of magnitude [23,24,25,26].

Energy changes concerning the segregation of chromium and nickel are relatively negligible compared to those of strongly surface-active elements such as oxygen. Therefore, they influence surface tension to a significantly smaller extent. In addition, nickel and chromium differ in their affinity for oxygen, which can affect the determination of surface tension in multicomponent systems [9,19,27,28,29,30,31,32].

Over the past two decades, several researchers have addressed the issue of wetting at the interface of Cr-Ni steel and refractory material under different oxygen partial pressure [33,34,35,36,37,38,39]. To the best of our knowledge, no one has addressed the effect of oxygen contained in the order of hundreds of ppm on the interfacial behavior of low-alloy steels containing chromium and nickel. Although low-alloy steels are widely used, mainly due to their unique properties, there is still a lack of experimental data in the literature regarding surface tension and interaction with refractory materials. This article contributes to understanding the surface and interfacial phenomena that occur at the interface of chromium and nickel steels with a higher oxygen content (240–900 ppm) and a corundum substrate.

## 2. Materials and Methods

### 2.1. Preparation of Steel Samples

In total, two series of steels, namely chromium (FeCOCr) and nickel (FeCONi), each containing three samples, are prepared. Pure metals (Fe, Cr, and Ni of 99.99% purity), carbon (99.99% purity), Fe_2_O_3_ tablets (99.999% purity), and other elements are used for their preparation. All precursors are melted in a vacuum induction furnace (Leybold Vacuum GmbH, Cologne, Germany). The melt is then cast into a vertically oriented mold where it solidifies into approximately 3 kg ingots, which are subsequently machined into 0.5 cm diameter rolls. The contents of the main alloying elements, chromium and nickel, were approximately the same percentage by weight as the other elements, except oxygen, which is weighed in minute amounts (see Table 1). Most elements are determined by GDOES (Glow Discharge Optical Emission Spectrometry) using a GDA-750 HP instrument supplied by SPECTRUMA GmbH (Bochum, Germany). Content of carbon, oxygen, and sulfur is measured on Eltra 2000 CS and 2000 ONH combustion analyzers (ELTRA, Haan, Germany).

### 2.2. Pretreatment of Samples

Before wetting tests, cylindrical steel specimens (5 mm diameter × 5 mm height) are thoroughly cleaned to remove surface oxides and sonicated in acetone. Furthermore, alumina plates (99.8% Al_2_O_3_) serving as substrates are annealed at 1150 °C for 6 h, and their surface is cleaned with acetone shortly before the experiment.

### 2.3. Determination of Interface Wetting

Surface tension and wetting angles are determined using a Clasic high-temperature observation furnace (CLASIC CZ s.r.o., Řevnice, Czech Republic) in which low-alloy steel specimens placed on corundum plates are inserted. Then, the furnace is hermetically sealed, evacuated to approximately 0.1 Pa, and purged with argon (purity greater than 99.9999%) for 30 min. The last two steps are repeated once more. The samples are heated from ambient temperature to 1400 °C at a heating rate of 5 °C·min^−1^. Consequently, the heating is slowed to 2 °C·min^−1^ until a final temperature of 1600 °C is reached. Contact times were the same for all tests, with the final stage from 1400 °C lasting 1 h and 40 min. The temperature is recorded with a Pt-13% Rh/Pt thermocouple near the sample. During heating, an inert atmosphere of argon is maintained in the furnace to prevent the oxidation of steel samples. Each sample is measured three times. The formation of steel droplets during heating is recorded with a CANON EOS550D camera. The acquired images are evaluated using the axisymmetric drop shape analysis (ADSA) method, which fits the drop profiles to a Laplacian curve using nonlinear regression analysis. This procedure allows determining the mentioned surface characteristics.

### 2.4. SEM, EDX and XRD Analyses

After high-temperature trials, scanning electron micrographs of steel droplets and corundum substrates are obtained by a JEOL 6490LV scanning electron microscope (JEOL Ltd., Tokyo, Japan) coupled with an INCA EDX analyzer (Oxford Instruments, Abingdon, UK), allowing quantitative analysis of microsized particles and determination of the chemical composition, to assess their interaction. SEM/EDX analyzes are performed with the following settings: thermoemission cathode LaB6, voltage 20 kV for high vacuum and 15 kV for low vacuum, high vacuum (10^−3^ Pa) for steel samples, and low vacuum (25 Pa) for corundum substrates.

Powder X-ray diffraction analyzes are carried out using a Bruker-AXS D8 Advance instrument (BRUKER, Germany) with a 2θ/θ measurement geometry and the positionally sensitive detector LynxEye under the following conditions: CuKα/Ni-filtered radiation, voltage 40 kV, current 40 mA, step mode with a step of 0.014° 2θ, total time 25 s per step and angular extent 5–80° 2θ. The data is processed by the Bruker AXS Diffrac and Bruker EVA software. The PDF-2 database (International Centre for Diffraction Data).

### 2.5. Differential Thermal Analysis Studies

The experimental measurement of liquidus temperatures is performed using a Setaram Setsys 18™ instrument (SETARAM Instrumentation Ltd., Lyon, France) equipped with S-type thermocouples. Steels are machined into cylindrical specimens (3.5 mm diameter, 3.1 mm height, and approximately 190 mg). Before the experiment, they are brushed, immersed in acetone, and placed in corundum crucibles. The device is evacuated and purged three times with argon gas (6N). During heating and cooling, the measuring chamber is flushed with argon at 2 L·min^−1^ to prevent oxidation of the steel samples. Liquidus temperatures are determined while heating at a rate of 10 °C·min^−1^. The measured values are corrected for the melting temperatures of high purity metals, Ni(5N) and Pd(5N), and the experimental conditions, i.e., the heating rate and the weight of the sample. Each steel sample is measured three times, and the average values are shown in Table 2.

## 3. Results and Discussion

### 3.1. Determination of Liquidus Temperatures

In addition to surface tension, wetting tests can also provide information on liquid temperatures, which are of practical importance for casting and solidification processes. This temperature was determined optically based on changes in sample silhouettes during heating [40] and by the DTA method, see Table 2. The experimentally determined temperatures were compared with the temperatures calculated using available software (FactSage 7.2, ThermoCalc 2019a, JMatPro 12.0, and IDS 1.3.1).

The calculated liquidus temperatures were more or less the same but showed differences from those obtained experimentally. The differences between experimental and theoretical temperatures will be discussed in Section 3.3.

### 3.2. Determination of Surface Characteristics

Surface tension and wetting angles were evaluated by an in-house application using the ADSA method in a temperature interval of interest, i.e., from the liquidus temperature obtained optically up to 1600 °C. In such a way, the temperature dependences of the surface tension and the average wetting angles were analyzed. Figure 1 shows the temperature dependence of the steel surface tension.

Chromium steel samples showed a moderate increase depending on temperature; in other words, the surface tension temperature coefficient was positive; see Table 3, in which fitted parameters are related to Equation (1).
(1)$$\sigma \left(T\right)=\sigma_{ref}+\frac{d\sigma }{dT}\cdot \left(T-T_{ref}\right)$$
where $\sigma_{ref}$ (mN·m^−1^) is surface tension at reference temperature $T_{ref}$ (°C), and $\frac{d\sigma }{dT}$ (mN·m^−1^·°C^−1^) denotes temperature coefficient of surface tension.

In the case of FeCONi steels, the coefficient was slightly negative or even constant. This can be attributed to the different sulfur contents of the steel series reported by [16,41,42], with a sulfur content around 55 ppm being considered critical to determining whether the trend in temperature dependence is positive or negative.

In addition to Figure 1, the oxygen content influenced the surface tension. As the oxygen content increased in minimal quantities, the surface tension of the steels investigated decreased significantly, which was also confirmed statistically by the Kruskal–Wallis test [43], according to which the effect was statistically significant (*p*-value << 0.001). Oxygen is a surface-active element segregating on alloy surfaces. The driving force of this phenomenon is the difference in surface energy and heat of solution between alloy constituent components, whereby elements with lower surface energy and positive heat of solution tend to segregate to the surface [9]. Here, at concentrations on the order of hundreds of ppm, oxygen forms a monolayer and changes the arrangement of the steel components compared to the bulk, which in effect reduces the surface tension [44]. The change in surface tension also had its response in wetting angles.

The wetting angles that characterize the degree of interaction between the steel and the substrate are provided in Figure 2. There was only a very slight dependence on the temperature in the case of both series. However, the oxygen content affected the values of the wetting angles, reducing them in both cases.

A significant difference in the contact angle values can be observed when comparing the chromium and nickel steel series, see Figure 2. While Ni-steels in contact with the corundum substrate achieved wetting angles in the range of 100 to 120°, Cr-steels showed much higher contact angle values ranging between 130 and 142°. These differences can also be seen in Figure 3. The wetting experiments proved a non-wetting behavior since the contact angle was higher than 90 deg., with nickel steels wetting the substrate more significantly.

### 3.3. Wetting Mechanism

To clarify the wetting mechanism, we performed several analyses. After wetting tests, the samples (steel/corundum substrate) of both series were subjected to SEM/EDX investigation (Figure 4 and Table 4).

For simplicity, we will only consider samples 2 and 5, which can be viewed as representative in terms of interaction. In the case of the chromium series, the free surface was locally covered with a fine layer of chromium oxide (Figure 4A) due to the higher oxygen and chromium content (approximately 4.5 wt%), which prevented the droplet from shaping in the early stages of melting. For this reason, the measured properties (surface tension, wetting angles) are only given from a temperature of 1550 °C. On the other hand, for the nickel series, a thin discontinuous layer of iron oxides was detected on the free surface of the droplet, which did not influence drop shape forming nor optical determination of the liquidus temperature (Figure 4E). It should be noted that the nickel series contained twice the amount of oxygen as the chromium series. The thickness of the oxide layer formed on the droplet surfaces reached 1.2–2.2 µm, and can be seen in Figure 4B,F. This layer was formed due to a larger amount of oxygen of the order of 10^2^ ppm. As confirmed by our previous studies, when using the same apparatus and conditions, no oxide layer was present on the surface when the oxygen content was around 10 ppm [45].

SEM images of the corundum substrate are shown in Figure 4C,D,G,H. In the case of specimen 2, the surface of the corundum substrate was partially melted in the wetted area, and iron and chromium particles were detected locally. In contrast, a nearly intact corundum structure and scattered iron particles were found in the outer region. Analysis of the corundum substrate of sample 5 gave similar results, that is, the area below the drop was locally melted and contained iron particles bearing trace amounts of nickel. The surrounding area remained unchanged except that iron particles were found on its surface. In addition, the average particle size was on the order of micrometers (2–5 µm).

The SEM/EDX microstructural characterization results suggest that the melting of the substrate and steel with the partial dissolution of the steel and subsequent precipitation process, without the formation of interfacial products between the steel and the substrate, are involved in the wetting mechanism. FactSage simulations were performed over a wide temperature range to support these findings. This approach is commonly used to complement wetting experiments [38,46], and the main benefits include energy and time savings. The following databases, FToxid, FSsteel, and FactPS, were selected for our calculations.

First, a simulation of the interactions that take place at the phase interface of sample 2 was performed (Figure 5). Up to a temperature of approximately 1520 °C, the FCC and BCC steel matrices were the dominant phases. They were then swiftly suppressed in favor of the liquid phase reaching a maximum at a temperature of 1529 °C. Also, the formation of various oxides, carbides, and sulfides occurred in a broad temperature range but to a much lesser extent. As regards the corundum substrate, the evolution of phases was more diversiform. The most abundant was corundum, whose declines were accompanied by the formation of other phases, i.e., calcium aluminosilicate and hibonite, beginning at temperatures of <500 °C and 1110 °C, respectively. In addition, other phases such as garnet and melilite also occurred, with the onset temperature of the former phase and the extinction temperature of the latter phase being the same, i.e., 735 °C. Of particular interest is the partial melting of the substrate surface at temperatures above 1425 °C which was suggestive of the proposed mechanism. The interaction of sample 4 was also simulated, with the same phases forming, but with the difference that their formation temperatures were slightly shifted, namely the substrate melting occurred at 1405 °C.

Second, we took a closer look at the free surface of the steels and the formation of oxides. The simulation was carried out in the temperature range 1000–1600 °C (Figure 6). In the case of chromium steel, only chromium oxide was formed, which was stable up to a temperature of 1585 °C. When the chromium content was reduced by three orders of magnitude, iron oxide and chromite (FeCr_2_O_4_) were produced. Reducing the oxygen content while maintaining the original chromium level resulted only in shortening the Cr_2_O_3_ occurrence interval. On the contrary, for nickel steel, iron oxide, manganese oxide, and chromite were present up to about 1500 °C.

Changes in the corundum substrate caused by temperature loading and interaction with steel samples were investigated by X-ray diffraction measurements made after wetting trials. In the case of the substrate of sample 2, there was a bifurcation of the corundum diffraction lines (Figure 7A), which is presented in more detail and at higher angles in Figure 7B, where the differences in the intensities and shapes of the lines, in other words, in peak half-widths after background subtraction, are well documented.

According to lattice parameters, the more pronounced phase can be interpreted as unsubstituted corundum, whereas the second one is Cr-doped corundum. Variation in half-widths indicates significant crystallite size differences, 907(30) nm for unsubstituted and 36.62(77) nm for substituted corundum. Assuming that the change in unit parameters depends linearly on the chromium oxide content, the measured lattice parameters a_0_ and c_0_, which were a_0_ = 4.77709(38) Å, c_0_ = 13.0598(13) Å fall within the interval corresponding to a 10% substitution by weight [47,48], where the uncertainties in parentheses were evaluated by the Rietveld optimization. It should be noted that Cr(III) ions can enter the corundum structure and substitute for Al(III) since they are only a bit larger. Chromium ions have two strong absorption bands in the visible spectrum, which give the substituted corundum a reddish color [49].

In the case of the nickel sample, alpha iron was detected, and no diffraction line splitting occurred. In both series, no other phases were present, which contradicts the findings of the FactSage simulation. The reasons may be multiple. Phase formation may have occurred at a concentration below the detection limit. Also, the simulations assume the homogeneity of the specimen. In our case, the corundum substrate consists of a dominant alumina phase in which calcium silicate microparticles are dispersed, and therefore the homogeneity condition is not met at the microscopic level.

## 4. Conclusions

The present experimental study assesses the effect of oxygen content (on the order of 10^2^ ppm) in FeCOCr and FeCONi steels on the determination of interface wetting by the sessile drop method. The results of this study can be summarized as follows:The surface tension of chromium low-alloy steels increases slightly with increasing temperature, while for nickel low-alloy steels, there is a decreasing trend in surface tension depending on temperature. A positive temperature coefficient of surface tension is usually associated with alloys having positive values of excess free energy. Sulfur, like oxygen, is a strongly surface-active element that segregates in the surface layer and reduces surface tension. However, as the temperature increases, sulfur desorbs into the bulk of the liquid metal, causing an increase in surface tension. This applies to alloys with a sulfur content greater than 50 ppm. In addition, in both series, oxygen reduced surface tension.The wetting angles between the investigated steels and the corundum substrate were almost temperature independent, and they decreased with increasing oxygen content. The contact angles of the nickel series were significantly smaller than those of the chromium one.In the case of the chromium series, the oxygen content complicates the determination of surface tension in the temperature interval from liquidus temperature to 1550 °C due to the formation of a Cr_2_O_3_ layer.The results of SEM and XRD analyses confirmed the nonreactive wetting. For the chromium series, chromium-doped corundum was found on the surface of the wetted area. The non-wetted surface remained utterly unchanged.

Taken together, this paper describes the interfacial phenomena between low-alloy steel having higher oxygen content and alumina refractory material, which is of use in various technological applications where melted alloys are in contact with ceramic containers and linings.

## Figures and Tables

**Figure 1 materials-15-01592-f001:**
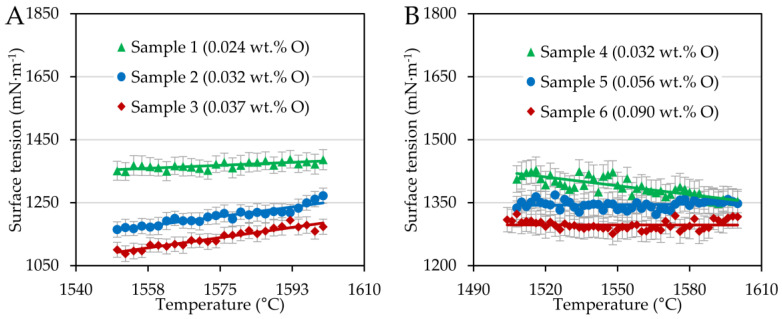
Surface tension plotted as a function of temperature for FeCOCr steel (**A**) and FeCONi (**B**) samples.

**Figure 2 materials-15-01592-f002:**
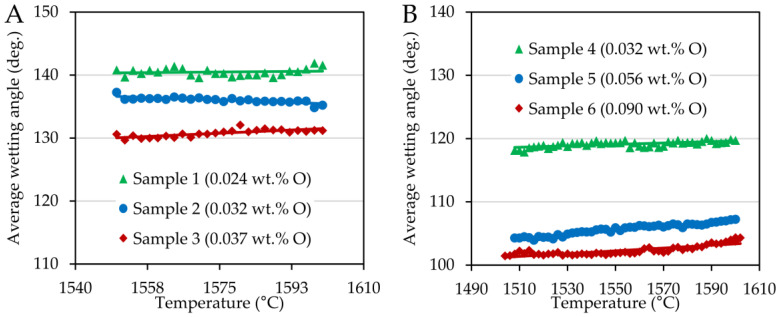
Average wetting angles plotted as a function of temperature for FeCOCr steel (**A**) and FeCONi (**B**) samples.

**Figure 3 materials-15-01592-f003:**
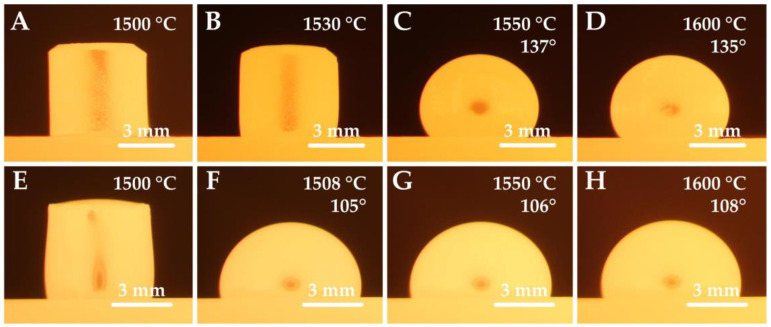
Steel silhouettes during thermal loading; sample 2 (**A**–**D**) and sample 5 (**E**–**H**).

**Figure 4 materials-15-01592-f004:**
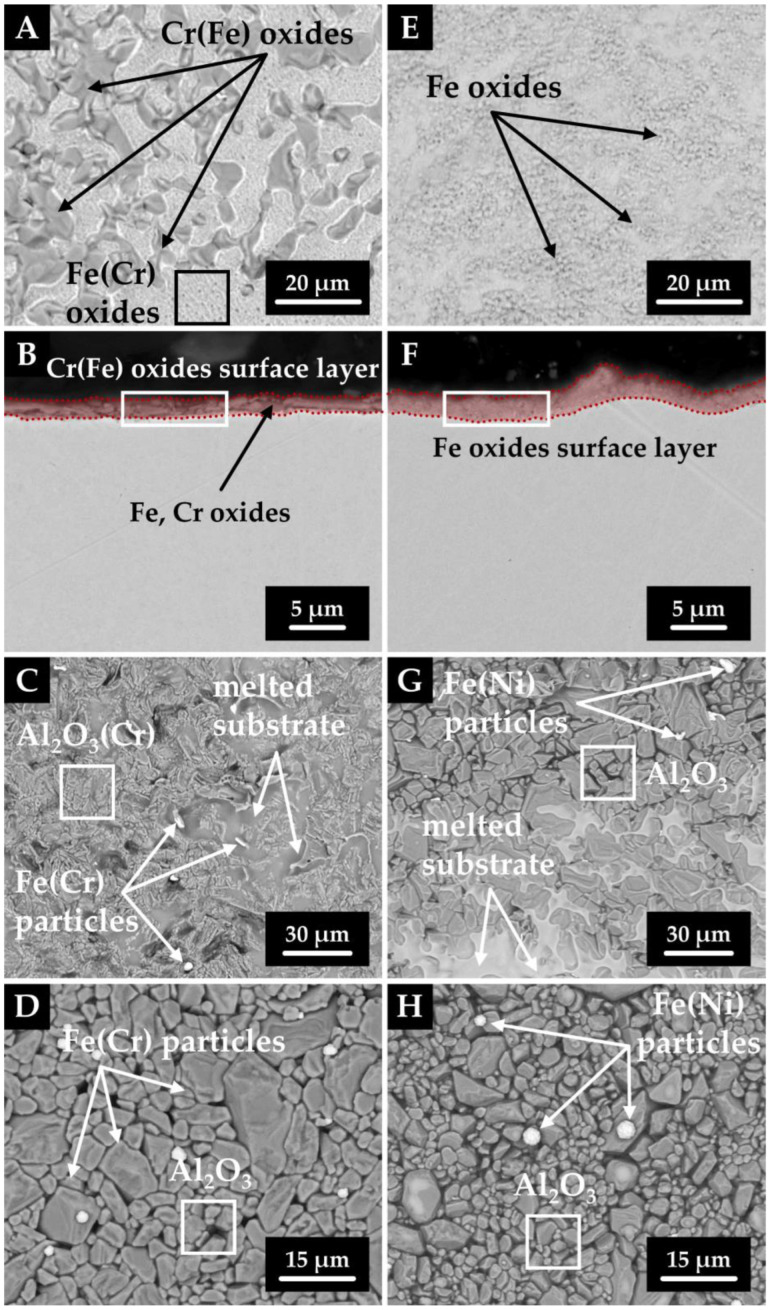
SEM images of sample 2 (left column) and 5 (right column) after wetting test, with rows from top to bottom showing: droplet free surface (**A**,**E**), droplet cross-section (**B**,**F**), wetted area of corundum substrate (**C**,**G**), and unwetted surroundings (**D**,**H**).

**Figure 5 materials-15-01592-f005:**
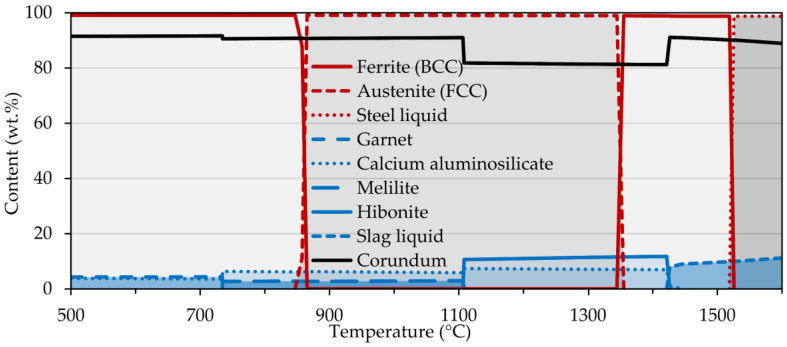
Interaction between corundum substrate and sample 2 calculated using FactSage.

**Figure 6 materials-15-01592-f006:**
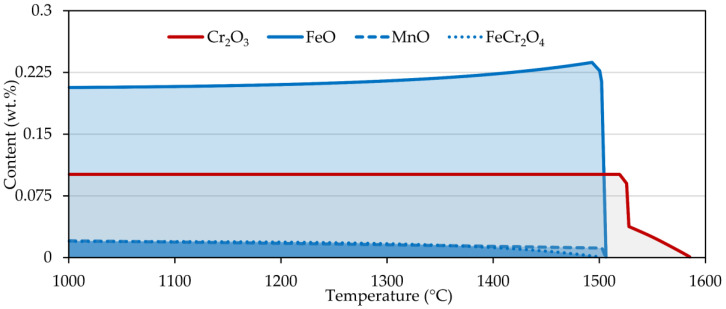
Oxide formation for sample 2 (red curve) and sample 4 (blue curve) calculated in FactSage 7.2.

**Figure 7 materials-15-01592-f007:**
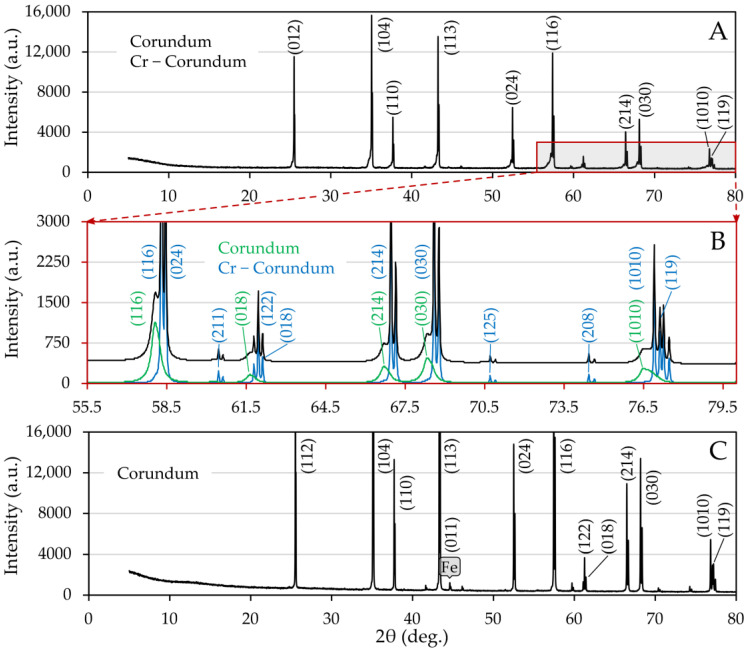
Indexed XRD patterns of corundum substrates, proving that there was almost no interaction between steel and corundum; (**A**) sample 2, (**B**) detail of diffractogram A showing a bifurcation of corundum diffraction lines obtained by Rietveld analysis, (**C**) sample 5.

**Table 1 materials-15-01592-t001:** Elemental composition of low-alloy steel samples.

Sample	O	Cr	Ni	C	Co	Mn	S	P	N
	(wt%)								
1	0.024	4.645	0.002	0.003	0.023	0.049	0.006	0.004	0.002
2	0.032	4.553	0.003	0.002	0.027	0.034	0.006	0.005	0.003
3	0.037	4.386	0.002	0.003	0.009	0.024	0.007	0.003	0.002
4	0.032	0.009	5.000	0.002	0.008	0.023	0.004	0.004	0.004
5	0.056	0.009	4.830	0.003	0.009	0.016	0.004	0.004	0.004
6	0.090	0.005	4.960	0.003	0.007	0.012	0.004	0.004	0.006

Content of other elements (Si, Al, Cu, Ti, Mo, V and B) was less than 10 ppm.

**Table 2 materials-15-01592-t002:** Comparison of liquidus temperatures determined either optically or by listed software applications.

Sample	Optical Method	DTA	FactSage	ThermoCalc	JMatPro	IDS
	(°C)					
1	1550	1527	1528	1529	1528	1530
2	1550	1527	1529	1529	1529	1530
3	1550	1528	1529	1530	1529	1530
4	1508	1516	1516	1518	1518	1517
5	1508	1514	1514	1518	1516	1518
6	1504	1513	1515	1518	1514	1517

All calculations were performed assuming equilibrium state. Elements that were not included in the calculations were: ThermoCalc—Co, S, P, N Si, Al, Cu, Ti, Mo, V, and B; IDS—N, S.

**Table 3 materials-15-01592-t003:** Results of the linear fit to calculate the linear dependence of the surface tension.

Sample	*T_ref_* (°C)	*σ_ref_* (mN·m^−1^)	*dσ/dT* (mN·m^−1^·°C^−1^)	Δ*T* (°C)
1	1550	1355	543.3 × 10^−3^	1550–1600
2	1550	1164	1717.9 × 10^−3^	1550–1600
3	1550	1096	1852.3 × 10^−3^	1550–1600
4	1508	1420	−695.0 × 10^−3^	1508–1600
5	1508	1344	36.2 × 10^−3^	1508–1600
6	1504	1297	−0.2 × 10^−3^	1504–1600

**Table 4 materials-15-01592-t004:** Results of semiquantitative EDX microanalyses.

Figure	Caption	O	Al	Si	Ca	Cr	Fe
		(wt%)					
4A	Cr(Fe) oxides	39.2	2.7	0.0	0.0	47.0	11.1
	Fe(Cr) oxides	23.9	0.5	0.4	0.0	4.1	71.1
4B	Cr(Fe) oxides surface layer	24.3	0.4	0.8	0.1	3.2	71.2
	Fe(Cr) particles	32.9	2.1	0.2	0	24.6	40.2
4C	Al_2_O_3_(Cr)	43.5	33.6	2.1	2.1	18.7	0.0
	Fe(Cr) particles	13.5	10.4	0.9	0.9	7.5	66.8
4D	Al_2_O_3_	47.0	50.0	1.0	0.9	0.9	0.2
	Fe(Cr) particles	27.4	18.5	0.3	0.4	3.8	49.6
4E	Fe oxides	22.4	0.5	0.2	0.0	3.8	73.1
4F	Fe oxides surface layer	15.9	0.1	0.3	0.0	4.4	79.3
4G	Al_2_O_3_	48.3	50.4	0.5	0.7	0.0	0.1
	Fe(Ni) particles	9.4	16.0	0.2	0.3	3.2	70.8
4H	Al_2_O_3_	43.6	50.9	2.7	2.6	0.1	0.1
	Fe(Ni) particles	32.3	16.2	0.2	0.2	3.3	47.8

For small particles and thin layers, the influence of the surroundings must be considered.

## Data Availability

The data presented in this study are available on request from the corresponding author.

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
