# Peer review of "The Effect of Trace Oxygen Addition on the Interface Behavior of Low-Alloy Steel"

_materials, 2022, doi:10.3390/ma15041592_

Round 1
Reviewer 1 Report
" Effect of Oxygen Added in Minute Quantities on Surface Properties and Interfacial Behavior of Low-alloy Steels "studied the surface and interfacial phenomena that occur at the interface of chromium and nickel steels with a higher oxygen content (240-900ppm) and corundum substrate, but minor modifications are still needed.
- The O, Cr and other elements of the test samples in Table 1 are different. Is there a lack of control group?
- Table 2 illustrates the sample temperatures tested by different methods, but does not explain which instrument is ultimately used as the benchmark. Although the temperature given in Table 3 seems to be obtained by using the optical method, why this method is selected is not explained, resulting in the redundancy of Table 2.
3. Fig. 1, Fig. 2 and Fig. 3 show the effects of different temperatures and oxygen contents on surface tension and wetting angle. However, the paper only describes the change trend and range, and does not explain the reasons. It is suggested to supplement the analysis of the figure.
Author Response
We thank the reviewer for reading our manuscript and for his comments, which we appreciated. We have revised the manuscript according to the recommendations and responded to the individual remarks.
- The O, Cr and other elements of the test samples in Table 1 are different. Is there a lack of control group?
Thank you very much for your question. Two series of samples with different oxygen contents (240 - 900 ppm) were prepared following the procedure described in Section 2.1. The first series (samples 1-3) contained approximately 5 % chromium, and the second series (samples 4-6) had about 5 % nickel. This study aimed to describe the effect of oxygen on the surface properties (surface tension and wetting angles) of low-alloy steels. Since the influence of other surface-active elements was either several orders of magnitude smaller (such as chromium, nickel, and other metals) or the differences in abundance were within 10 ppm, the differences in composition, except for oxygen content, can be neglected. In addition, it is complicated to control the content of all elements in such a complex system mainly due to the efficiency of Fe2O3 dissolution, the interaction of oxygen with other elements leading to slag formation, etc. Therefore, the samples were repeatedly prepared in relatively large quantities (3 kg ingots) and at high temperatures to minimize these effects.
- Table 2 illustrates the sample temperatures tested by different methods but does not explain which instrument is ultimately used as the benchmark. Although the temperature given in Table 3 seems to be obtained by using the optical method, why this method is selected is not explained, resulting in the redundancy of Table 2.
Thank you very much for your notice. Table 2 compares liquidus temperatures determined both experimentally (DTA, optical method) and theoretically using available software (FactSage 7.2, ThermoCalc 2019a, JMatPro 12.0, and IDS 1.3.1). It emphasizes the difference in the liquid temperatures determined by the optical method, especially for the chromium series of steel samples. Using this method, the liquidus temperature is attributed to the temperature at which the droplet forms. The issues associated with the delayed droplet formation of this series are discussed in Section 3.3. The data in Table 3 are related to the evaluation of the temperature dependence of the surface tension. Surface tension can only be determined after the steel sample is „packed” into a droplet. Thus, the temperature given in Table 3 agrees with the liquidus temperature determined by the optical method. The term reference temperature given in equation (1) is commonly used and means the temperature that usually corresponds to the liquidus temperature, but not in the case of the chromium series.
Temperatures obtained by DTA can be used as reference values for the following reasons.
- very high sensitivity in determining temperature changes during phase transitions due to the configuration of the measuring system,
- homogeneous temperature field of the device,
- sophisticated methodology regarding "raw temperature" corrections based on heating rate, sample weight, and melting of pure standard (Pd5N and Ni5N),
- vast experience with measurement of phase transition temperatures and other properties by DSC and DTA method ("Applied Surface Science 2021, 552: 149490","Journal of Alloys and Compounds 2021, 865: 158715", "Applied Surface Science 2020, 529: 147086", "Journal of Thermal Analysis and Calorimetry 2020, 142(2): 535–546", "Applied Clay Science 2019, 179: 105150","Journal of Materials Research and Technology 2019, 8(4): 3635–3643","Journal of Thermal Analysis and Calorimetry 2017, 127(1): 123–128","Journal of Thermal Analysis and Calorimetry 2017, 127(1): 423–429").
- 1, Fig. 2 and Fig. 3 show the effects of different temperatures and oxygen contents on surface tension and wetting angle. However, the paper only describes the change trend and range, and does not explain the reasons. It is suggested to supplement the analysis of the figure.
Thank you very much for your recommendation. We have discussed the mechanism in the Results and Discussion section (see lines 196 ‒ 203).

Reviewer 2 Report
- For the title, surface property is an ambiguous term, and it is suggested to change it to surface tension. The implications of surface tension are covered by interface behavior and should be removed. Therefore, the title can be changed to: The effect of trace oxygen addition on the interface behavior of low alloy steel.
- For the abstract, line 21-22, all descriptions of the software can be deleted, and it is recommended to change the temperature to the theoretically calculated temperature value. The specific calculation method can be described in detail in the corresponding part of the text.
- Why does the addition of trace oxygen reduce the surface tension and wetting angle? The corresponding physical mechanisms should be supplemented in the abstract.
- In line 42 and line 50, the citation is unreasonable. Never cite all references at the end of a paragraph, but cite references separately after the corresponding sentence.
- ‘Surface properties’ is a highly generalized and broad term that includes not only surface tension and wetting as listed by the authors, but also catalytic activity, biological activity (inertness), adsorption, corrosion, etc. Therefore, it is recommended that the authors change the surface properties to interface wetting.
- The title of Table 1 is inappropriate. The test methods for C, O, and S described herein are not the GDA 750 HP Optical Emission Spectrometer as defined in the title of Table 1.
- In line 125, the vacuum degree of SEM is only 10-3 Pa. The authors must verify this data carefully. To my knowledge, most SEMs are at least 10-5 Pa.
- There are many grammatical errors in the text. The description of the experimental results should use the simple present tense.
- It is well known that the contact angle between the metal liquid and the substrate is related to the contact time. How long is the contact time for the wetting angle values at different temperatures obtained in the article? Are contact times the same for all tests? These should be stated in the experimental methods.
- All figures and tables should be typeset to the right, not centered.
- For Figure 7, all diffraction peaks should be indexed. Also, are there two XRD curves in Figure 7a? The reader cannot visually tell how many curves there are in the graph. Inexplicably, 3 more curves appeared in Figure 7b. This is unreasonable, and the author must express it clearly.
- Line 276, 907 (30) nm, what does the number (30) in parentheses mean? A brief description should be given in the text.
- From the experimental results, the surface tension of chromium low alloy steel slightly increases with the increase of temperature, while the surface tension of nickel low alloy steel tends to decrease with the increase of temperature, which seems to be related to the two facts:1) formation of Cr2O3 layer and 2) the diffusion of chromium to the corundum substrate. However, in the main text and the conclusion, the author did not give a clear indication and explanation, which needs to be supplemented.
Author Response
We thank the reviewer for the careful reading of the manuscript and providing us with constructive remark and comments which we have considered. Please, find below a detailed point-by-point response to them.
- For the title, surface property is an ambiguous term, and it is suggested to change it to surface tension. The implications of surface tension are covered by interface behavior and should be removed. Therefore, the title can be changed to: The effect of trace oxygen addition on the interface behavior of low alloy steel.
Thank you very much for your advice, which we appreciate. We have changed the title according to your recommendation.
- For the abstract, line 21-22, all descriptions of the software can be deleted, and it is recommended to change the temperature to the theoretically calculated temperature value. The specific calculation method can be described in detail in the corresponding part of the text.
Thank you very much for your recommendation. We rewrote mentioned sentence (see line 22).
- Why does the addition of trace oxygen reduce the surface tension and wetting angle? The corresponding physical mechanisms should be supplemented in the abstract.
Thank you very much for your question. In the abstract (see lines 15 ‒17), we have only briefly mentioned why oxygen reduces the surface tension due to the size limitation of the abstract. We have discussed the mechanism further in the Results and Discussion section (see lines 196 ‒ 203).
- In line 42 and line 50, the citation is unreasonable. Never cite all references at the end of a paragraph, but cite references separately after the corresponding sentence.
Thank you very much for your notice. We have corrected the citation according to the recommendation.
- ‘Surface properties’ is a highly generalized and broad term that includes not only surface tension and wetting as listed by the authors, but also catalytic activity, biological activity (inertness), adsorption, corrosion, etc. Therefore, it is recommended that the authors change the surface properties to interface wetting.
Thank you very much for your recommendation. We have made the appropriate changes to the manuscript.
- The title of Table 1 is inappropriate. The test methods for C, O, and S described herein are not the GDA 750 HP Optical Emission Spectrometer as defined in the title of Table 1.
Thank you very much for your notice. The table caption has been changed accordingly.
- In line 125, the vacuum degree of SEM is only 10-3 The authors must verify this data carefully. To my knowledge, most SEMs are at least 10-5 Pa.
Thank you very much for your notice. We checked the pressure readings. The pressure in the specimen chamber was 10-3 Pa, and the pressure inside of the electron optical system was 10-5 Pa. In addition, the pressure data for LV and HV modes were added.
- There are many grammatical errors in the text. The description of the experimental results should use the simple present tense.
Thank you very much for your recommendation. The manuscript has been checked by a native speaker Mark Landry (staff member). We rewrote the experimental section in the simple present tense.
- It is well known that the contact angle between the metal liquid and the substrate is related to the contact time. How long is the contact time for the wetting angle values at different temperatures obtained in the article? Are contact times the same for all tests? These should be stated in the experimental methods.
Thank you very much for your questions. The contact times were the same for all steels. The most important part of the heating was the final 100 minutes, which is mentioned in the revised manuscript (see lines 119 ‒ 120).
- All figures and tables should be typeset to the right, not centered.
Thank you very much for your notice. The tables and graphs were correctly aligned.
- For Figure 7, all diffraction peaks should be indexed. Also, are there two XRD curves in Figure 7a? The reader cannot visually tell how many curves there are in the graph. Inexplicably, 3 more curves appeared in Figure 7b. This is unreasonable, and the author must express it clearly.
Thank you very much for your remarks. X-ray diffraction patterns were indexed. In Figure 7a, there is only one curve. In Figure 7b, the black curve above is a diffraction measurement (corresponding to the record in Figure 7a), and the green and blue curves were obtained by Rietveld analysis and represent the contributions of the individual phases (corundum and Cr – doped corundum) to the record of the diffraction pattern.
- Line 276, 907 (30) nm, what does the number (30) in parentheses mean? A brief description should be given in the text.
Thank you very much for your recommendation. The number in parentheses corresponds to the standard uncertainty, and the description was added to the manuscript (see lines 306 ‒ 307).
- From the experimental results, the surface tension of chromium low alloy steel slightly increases with the increase of temperature, while the surface tension of nickel low alloy steel tends to decrease with the increase of temperature, which seems to be related to the two facts:1) formation of Cr2O3 layer and 2) the diffusion of chromium to the corundum substrate. However, in the main text and the conclusion, the author did not give a clear indication and explanation, which needs to be supplemented.
We thank the reviewer for this comment. This matter has been explained in the first paragraph of the conclusion.

Reviewer 3 Report
This work has the good intention to correlate the oxygen and alloying elements contents between the surface properties and interfacial behaviour of low-alloy steel. However, there are several issues that need to clarify.
- The design of the experiments: each sample has different oxygen content and different element content. Therefore, it is tough to attribute the reason for the variation of the wetting angles.
- Why was corundum chosen as the substrate? Will it affect the wetting angle if it melts in the wetting area? And it is very hard to tell from the SEM images if it melted or not.
- Why sample 2 and sample 5 were chosen to compare? They do not have the same oxygen content or any other element contents.
- Please include the EDX results alongside the SEM images. Without EDX results, It is impossible to tell what the arrows in Figure 4 pointed at.
Author Response
We thank the reviewer for the careful reading of the manuscript and providing us with constructive remarks. Please, find below a detailed point-by-point response to them.
- The design of the experiments: each sample has different oxygen content and different element content. Therefore, it is tough to attribute the reason for the variation of the wetting angles.
Thank you very much for your notice. Two series of samples with different oxygen contents were prepared according to the procedure described in Section 2.1. Since the influence of other surface-active elements was either several orders of magnitude smaller (such as chromium, nickel, and other metals) or the differences in abundance were within 10 ppm, the differences in composition, except for oxygen content, can be neglected. In addition, it is challenging to regulate the content of all elements in a poly-component system mainly due to the efficiency of Fe2O3 dissolution, the interaction of oxygen with other elements leading to slag formation, etc. Therefore, the samples were repeatedly prepared in relatively large quantities (3 kg ingots) and at high temperatures to minimize these effects.
- Why was corundum chosen as the substrate? Will it affect the wetting angle if it melts in the wetting area? And it is very hard to tell from the SEM images if it melted or not.
Thank you very much for your comment. We have used high-purity corundum, as it is almost noninteractive under given conditions, thus minimizing contamination of the steel, which can affect surface tension and wetting angles. Unfortunately, partial localized melting of the corundum substrate cannot be avoided, as it is thermodynamically favorable at such high temperatures. However, corundum substrates are commonly used in the sessile-drop method. In Figures 4 D and G, the melted areas of the corundum substrate have been marked, and in addition, Figure 4G has been replaced by another one in which the melting is clearly visible.
- Why sample 2 and sample 5 were chosen to compare? They do not have the same oxygen content or any other element contents.
Thank you very much for your question. Samples 2 and 5 were selected as average representatives of the chromium and nickel steel series with the median value of the oxygen content. The remaining two samples (1 and 3, 4 and 6) had similar results in SEM/EDX and XRD analyses and FactSage simulations within the given series. By selecting samples 2 and 5, we highlighted the different behavior (interactive wetting) of chromium and nickel steels at high temperatures.
- Please include the EDX results alongside the SEM images. Without EDX results, It is impossible to tell what the arrows in Figure 4 pointed at.
Thank you very much for your recommendation. Table 4 showing the results of semiquantitative EDX microanalyses has been added to the manuscript.

Reviewer 4 Report
The author works aims to assess the effect of an oxygen content graded in minimal quantities, on the order of thousands of ppms, on the determination of surface properties of low-alloy FeCOCr and FeCONi steels in contact with a corundum substrate. The sessile drop method was used for wetting tests in the temperature range from steel liquidus temperatures to 1600 °C under non-oxidizing conditions. The effect of oxygen on surface tension and wetting angles was verified by statistical analysis using the Kruskal-Wallis test, which supported the results stating that the values of these quantities decreased with increasing oxygen content. Further, liquidus temperatures, which are of practical importance, were determined by the optical and DTA methods and then compared with temperatures calculated using available software (ThermoCalc 2019a, FactSage 7.2, IDS 1.3.1 and JMatPro 12.0). It turned out that the increased chromium content causes difficulties in deter mining surface properties up to 1550 °C due to the formation of a thin Cr2O3 layer. In addition, SEM and XRD analyses accompanied by calculations in the FactSage oxide database were performed to understand the wetting mechanism better.
Find below some aspects that must be addressed by authors:
- The authors must clearly state the novelty and main contributions of this work when compared with a large amount of literature available over the last decades.
- In fig no 4, author could have mention the size of the particle (Fe, Ni) if possible
- Define in fig 7 (A), the grey colour box in right side down corner indicating what phenomenon?
- Comment on the experimental accuracy and efficiency of the proposed approach.
- The results are sound; anyway, it would be great if the author can clarify the benefits of the presented approach when compared with some alternative model/software.
- The “Conclusions” section must be expanded.
- Need to improve figure resolution.
- Kindly improve the English of the article.
Final comment: Major revision.
Author Response
We want to thank the reviewer for reading our manuscript and for invaluable feedback. We have carefully read the comments and revised the manuscript accordingly. Please, see below our responses.
- The authors must clearly state the novelty and main contributions of this work when compared with a large amount of literature available over the last decades.
Thank you very much for your comment. We have added the required information to the last paragraph of the introduction section.
- In fig no 4, author could have mentioned the size of the particle (Fe, Ni) if possible.
Thank you very much for your comment. Information on the particle size was added to the manuscript (see line 251).
- Define in fig 7 (A), the grey colour box in right side down corner indicating what phenomenon?
Thank you very much for your question. The grey rectangle corresponds to Figure 7B. Figure 7B has now been framed in red.
- Comment on the experimental accuracy and efficiency of the proposed approach.
Thank you very much for your comment. The method's accuracy is usually around 2 % but can be even greater. One factor that accounts for this is the contamination of steel samples due to the chemical interaction of the melt with alumina substrate. To minimize or avoid this, we used a high-purity corundum substrate. Besides, our software uses an advanced technique for evaluating droplet silhouettes. The experimental scatter is given by error bars (Figure 1).
- The results are sound; anyway, it would be great if the author can clarify the benefits of the presented approach when compared with some alternative model/software.
Thank you very much for your comment. In the manuscript, we have briefly highlighted the advantage of this approach (see lines 257 ‒ 260).
- The “Conclusions” section must be expanded.
Thank you very much for your comment. The “Conclusions” section was expanded in the revised manuscript.
- Need to improve figure resolution.
All figures were exported in a high resolution 1200 dpi and provided TIFF format as a single zip archive.
- Kindly improve the English of the article.
Thank you very much for your recommendation. The manuscript has been checked by a native speaker Mark Landry (staff member).

Round 2
Reviewer 2 Report
Questions raised by the reviewers have been well answered and the revised manuscript is ready for publication.
Reviewer 4 Report
I am satisfied with the response. Accept it